# A convenient five-segment cassette procedure for DNA insertions coding for novel peptides

Jonathan Filley🆔*

Oligometrics, Inc., Boulder, Colorado, United States of America

* jfilley@oligometrics.com

## Abstract

A DNA cassette assembly method is described which utilizes inexpensive oligomers no longer than 40 nt in length. The five-segment cassettes have 20 nt overlaps which give an effective length of 80 nt, making it possible to code for peptides up to 20 amino acids long. The cassettes have three phosphate free nicks, which can be successfully inserted into plasmid DNA and used to transform *E. coli*. The nicks are repaired *in vivo* by an unknown mechanism. Insertions are not successful for cassettes with greater than three nicks. A procedure is provided for rapid turnaround from DNA design to peptides, which are easily isolated as C-terminal fusions with GFP. The technique generally gives the expected sequence, with errors which occur about 1% of the time. Several representative DNA inserts are described which illustrate the method, as well as chemical details on the new peptides coded for. The peptides can be readily mutated to make it possible to understand how polar and aromatic residues affect GFP-fusion solubility, and how histidine residues can be strategically placed in a peptide for good IMAC retention. The method can be used to explore a large number of new designed peptides as fusion products quickly and economically.

## Introduction

Peptides can be made using solid phase support technology based on the original Merrifield method [1], genetic engineering [2], or by combinatorial techniques such as phage display [3]. The first two methods deliver material with the expected sequence, while the third allows for discovery of novel peptides, whose sequences, once known, can be used in the first two methods for the synthesis of actual peptides. For the peptide chemist wishing to explore novel sequences designed for specific functions, it is probably safe to say that a genetic engineering technique will be fastest and most cost effective for bringing a new and relatively long peptide sequence of up to 20 amino acids into existence, since the suite of methods, commercially available kits and reagents, and published procedures pertaining to genetic engineering, is very large. However, genetically engineered peptides synthesized *in vivo* encounter several challenges, primary among them being that short peptides are often degraded by the host organism before they can be isolated. A useful workaround to this problem is to express a peptide as a

**Data Availability Statement:** All relevant data are within the article.

**Funding:** The author(s) received no specific funding for this work.

**Competing interests:** The author has declared that no competing interests exist.

fusion product with a larger protein [4, 5]. In this case, a sequence that can be conveniently cleaved is engineered into the peptide such that the peptide can be separated from its fusion partner, and then separately purified. Alternatively, the cleavage step can be omitted, but now of course the possibility the larger protein can influence the chemistry of the peptide, or the peptide influence the protein, must be considered [6, 7]. Certain benefits of this non-cleavage strategy are speed and convenience, since fewer steps are involved in purification, and the large protein can have properties conducive to analysis (e.g. fluorescence).

One of the advantages of a robust genetic engineering economy is the availability of low cost DNA primers. Currently, DNA oligonucleotides up to 40 nt long are available at a cost of around $7. Since the oligomer sequence is under the complete control of the investigator, this opens up a huge number of possible peptides, where the DNA codes for the desired peptide, and includes restriction sites for the insertion of the DNA into plasmids with the appropriate cloning sites. Larger peptides can of course be coded by longer oligonucleotides, but now the cost is much higher, and in fact increases on a per-nucleotide basis. The longest commercially available custom DNA oligonucleotide is currently100 nt (~$46). It is worth mentioning these costs double for the double stranded DNA required for insertions.

Taking into account the nucleotides needed for restriction sites, a DNA 40-mer can code for a peptide only seven amino acids long. If the DNA length is doubled, the peptide can be 20 amino acids in length (see below). A project aimed at the synthesis of novel 20 amino acid peptides will in general require two DNA strands for each peptide, and two more strands for every mutation to the peptide that is contemplated. For DNA 80-mers, these costs can become significant as the number of mutations mount. However, we have found that an 80 nt cassette can be broken into five segments, each 40 nt in length or less, which reduces the cost significantly. Moreover, peptide mutations, if they occur in an "inter-segment" fashion, now cost $14, and make it possible to reuse segment DNA solutions multiple times as long they do not code for a mutation. Herein we wish to describe this five-segment cassette insertion method, and the phosphate-free nicks required for its implementation. Interestingly, the kinase reaction that would normally be necessary to make ligation of these nicks possible is not required; instead, the newly transformed bacteria seal the nicks as they replicate the plasmids.

## Materials and methods

All molecular biology experiments were carried out with *E. coli* HB101. Enzymes, desalt purity DNA, and DNA purification kits were obtained from Invitrogen. DNA sequencing was performed by TACGen. The pGLO plasmid (Bio-Rad) was cut with *SacI* to remove the stop codon and the codons for the terminal Tyr-Lys from the Green Fluorescent Protein (GFP) gene, leaving random DNA coding for a C-terminal peptide which has the effect of producing a phenotype with dim fluorescence, easily distinguished from the normal bright phenotype. This procedure delivers a vector for the easy insertion of cassettes with *SacI/XbaI* sites, where a stop codon is designed into the DNA coding for a peptide. Thus, all peptides made using this methodology are C-terminal fusion peptides with cycle-3 GFP, absent the terminal Tyr-Lys of GFP. Transformations were performed by heat shock and plasmid DNA was isolated by standard methods [8]. In a typical procedure, the five DNA segments dissolved in nuclease-free water but otherwise used as received, are mixed to produce double stranded DNA which is digested with *SacI* and *XbaI*. Simultaneously, the vector described above is digested with the same enzymes. After enzyme denaturation by heating to 65˚C 20 m, the insert is diluted 25-fold and added to the digested plasmid for ligation with T4 ligase, $[\text{plasmid}]_f \sim 5$ nM, $[\text{insert}]_f \sim 10$ nM. After heat shock, cells are streaked onto agar containing ampicillin and arabinose (the vector codes for beta-lactamase and the GFP gene is under the control of the

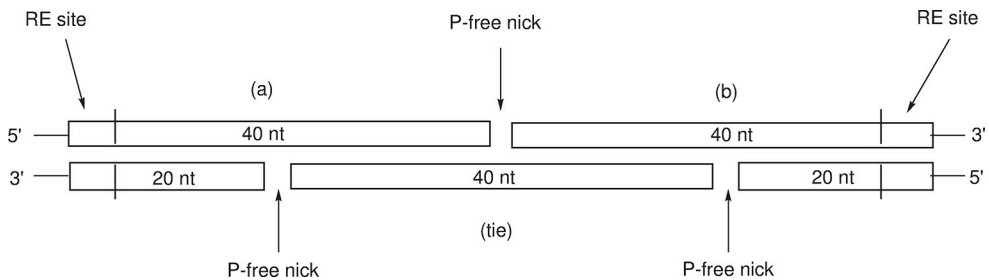

**Fig 1. Schematic diagram of a five-segment 80 nt cassette.** RE = restriction enzyme.

arabinose operon promoter $P_{BAD}$). Dim phenotypes are judged to be from cells harboring unchanged plasmid, while bright phenotypes with successful insertions are isolated and grown for further DNA and protein analyses. A diagram of the 5-segment cassette is shown in Fig 1. A preliminary report of this insertion technique has previously appeared [9].

Proteins were isolated from 20 ml cultures (LB broth, 0.25 mg/ml ampicillin, 2.5 mg/ml arabinose) grown overnight at 26°C from colonies transformed with the appropriate plasmid. Cells were pelleted, suspended in 5 ml TE buffer, treated with lysozyme (1.4 mg/ml) and frozen at -5°C. The thawed mixture was centrifuged to remove cell debris, and the supernatant carried forward for purification on hydrophobic interaction chromatography (HIC) columns (Bio-Rad Macro Prep Methyl HIC). The three-step procedure involves: 1) Six 5x9 mm spin columns (Invitrogen PureLink) equilibrated in 1.1 potassium sodium tartrate (Rochelle salt, RS) pH 8 onto which is loaded the supernatant diluted with an equal volume of 2.2 M RS, washed 1x with 300 μl 1.1 M RS and the fluorescent material is eluted with 500 μl 10 mM HEPES, pH 8; 2) Four 30x9 mm gravity columns equilibrated with 2 M ammonium sulfate (AS) onto which is loaded the eluate from step 1 diluted with an equal volume of 4 M AS and washed successively with 500 μl 1.3 M AS, 500 μl 0.7 M AS, and 500 μl 0.35 M AS (all at pH 8) before the fluorescent material is eluted with TE buffer; 3) Four 10x9 mm gravity columns equilibrated with 1.1 M RS onto which is loaded the eluate from step 2 diluted with an equal volume of 2.2 M RS and washed with 2x300 μl 1.1 M RS before the fluorescent material is eluted with 10 mM HEPES pH 8, final volume ~2 ml. Proteins are judged to be ~50% pure based on 280nm/395nm ratios [10–12] and SDS-PAGE results with yields of 0.1–0.3 mg. Immobilized metal affinity chromatography (IMAC) was performed on 12x9 mm columns using IMAC-Select Affinity Gel (Sigma-Aldrich) which had been converted to its $Ni^{2+}$ form using $Ni(OAc)_2$ and equilibrated with 20 mM phosphate pH 7.2. Proteins were analyzed for their IMAC affinity by washing the column with this buffer in 600 μl fractions and imaging them with UV light (Fig 2). The buffer was changed to 100 mM imidazole pH 8 at fraction 6 to elute fusions with high IMAC affinity.

## Results and discussion

This procedure reliably gives transformed colonies with two phenotypes, since two plasmids are present during heat shock: dim colonies with the re-formed starting plasmid with no new insert, and bright colonies with the recombinant plasmid harboring the newly designed DNA. Selection of the recombinant DNA is easily accomplished by picking bright colonies. This has been proven to be true for the majority of experiments, although it has been observed that certain peptide tails can interfere with GFP folding, in which case very dim colonies or even non-fluorescent colonies can be seen. In any case, a two step process has been proven to be effective, where the selected colony is first cultured overnight and its plasmid DNA is extracted for a

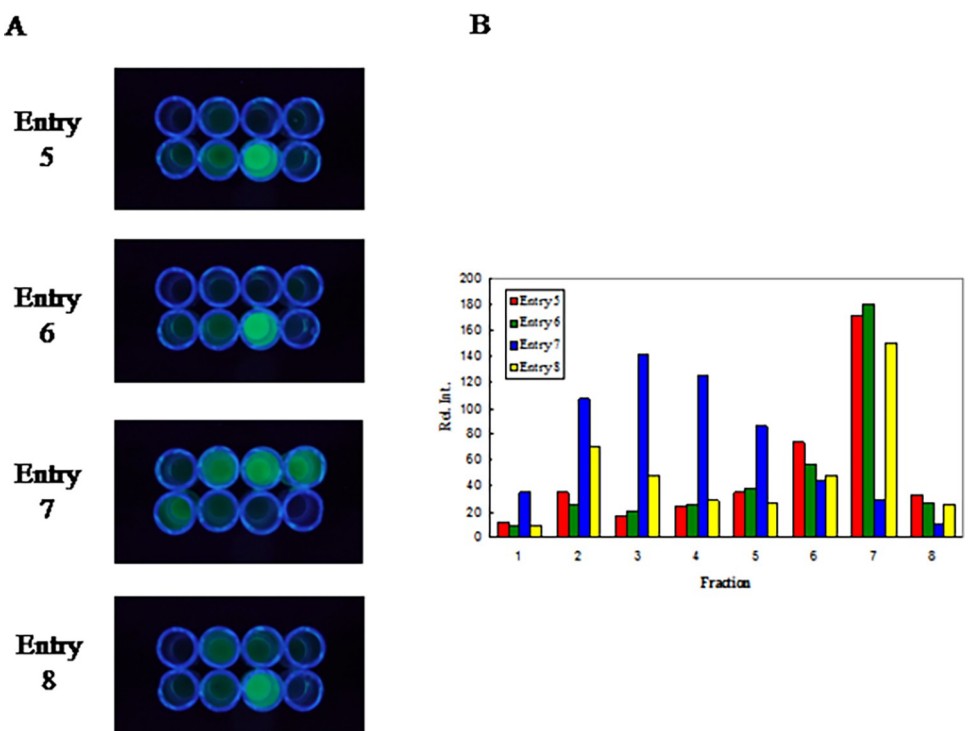

**Fig 2. Images and intensities of diluted GFP-peptide fusions used in this work.** A: Fractions 1–4 top L-to-R, 5–8 bottom L-to-R, after IMAC separation viewed with a transilluminator at 365 nm. Protein concentrations not held constant from run to run. B: Intensities obtained using ImageJ.

second round of transformation and colony growth. In this step, the large number of colonies often facilitates the confirmation of the phenotype. Culturing of a selected colony in this round can be scaled for kit-DNA isolation, suitable for sequencing. This protocol gives a rapid turnaround starting from DNA design to kit-DNA isolation which can also accommodate a larger scale protein culture, where the chemical properties of the GFP-peptide fusion (see below) often verify the identity of the insert, even before the DNA is sequenced.

The two step procedure described has been used for over 200 different DNA designs, with sequencing data confirming the identity of the recombinant plasmids. Table 1 displays eight sets of DNA which are representative of the inserts used. With a fast and reliable method for converting DNA designs into peptides, it is possible to test a number of ideas, some of which are more chemical in nature than biological. For example, questions can be asked about how the solubility of GFP is changed as the tail amino acids attached to it become more hydrophobic, and which hydrophobic amino acids have the biggest effect. Second, shorter alternatives to the traditional His$_6$ affinity tag can be explored, with the goal of determining if more than two histidine moieties is overkill when coordinating a metal during IMAC. And finally, peptides can be designed to fold to bring together two His residues which are far apart in the peptide sequence such that they can coordinate a metal and impart high IMAC affinity to the peptide (and its fusion partner), in an attempt to further understand the factors that control peptide folding. The peptides shown in Table 2 were designed with these questions in mind, and are coded by the DNA in Table 1. While the GFP-peptide fusions are only 50% pure, they are suitable for the experiments described here. The final washes with Rochelle salt (step 3 above) ensure the ammonium concentration is reduced to levels low enough to not interfere with the

**Table 1. Representative inserts used in this work.** Restriction sites are underlined and mutations are shown in bold. Segments "a" and "b" are shown in Fig 1. Complementary 3'-to-5' DNA is not displayed.

| Entry | Segment | Sequence |
|---|---|---|
| 1 | a | CCGAGCTCTGGTTCTTCCACCACTACTACTGGTGGTGGCA |
|   | b | CCACCACTTCTTCTGGTAACCCACGAGACCAATCTAGAGG |
| 2 | a | CCGAGCTCTGGTTCTTCCACGAGCACTACTACTGGGAGTG |
|   | b | GTGGCACCACGAGCACTTCTTCTGGGAGTAAGTCTAGAGG |
| 3 | a | CCGAGCTCTGG**TTCTTC**GACGAG**CACTACTAC**GACGAGTG |
|   | b | GTGG**CAC**GACGAG**CACTTCTTC**GACGAGTAAGTCTAGAGG |
| 4 | a | CCGAGCTCTGG**TGGTGG**GACGAG**TGGTGGTGG**GACGAGTG |
|   | b | GTGG**TGG**GACGAG**TGGTGGTGG**GACGAGTAAGTCTAGAGG |
| 5 | a | CCGAGCTCGGAGGTGGAGGTGGAGGTTATGGCCCG**ATT**CC |
|   | b | GTTTGGCCGCGGCCATGGCCATGGCCAGTAACTCTAGAGG |
| 6 | a | CCGAGCTCGGAGGTGGAGGTGGAGGTTATGGCCCG**GCG**CC |
|   | b | GTTTGGCCGCGGCCATGGCCATGGCCAGTAACTCTAGAGG |
| 7 | a | CCGAGCTCGGAGGTGGAGGTGGAGGTGGACATTATAGCGG |
|   | b | C**AAC**CCGCCGCCGGGCAGCGGCTGGCATTAACTCTAGAGG |
| 8 | a | CCGAGCTCGGAGGTGGAGGTGGAGGTGGACATTATAGCGG |
|   | b | C**AGC**CCGCCGCCGGGCAGCGGCTGGCATTAACTCTAGAGG |

IMAC runs. Clearly, for tails with high IMAC affinity, greater purity levels can be achieved if high purity preparations are required.

The Entries in Tables 1 and 2 serve to illustrate the advantages of the five segment insertion method as well as some of the peptide attributes that can be examined. For peptide Entries 1,2 and 3, the motivation was to increase solubility by adding polar carboxylate side chains to non-polar aromatic segments. Solubility in these experiments is judged by the presence of fluorescence in the supernatant (soluble) or in the cell debris pellet (insoluble). Thus, the string of random aromatics in Entry 1 was not made soluble with four carboxylates at the end of the chain, and solubility was not improved by interspersing the carboxylates along the chain in Entry 2. However, good solubility was achieved by doubling the carboxylates in Entry 3. Replacing all the aromatics in Entry 3 with tryptophan in Entry 4 once again renders the fusion insoluble, suggesting that close to eight carboxylate residues are required for good solubility when designing 20 amino acid tails appended to GFP. Entry 4 also highlights the potency of tryptophan in rendering the fusion insoluble. Note that if tryptophan in Entry 4 is replaced by tyrosine (sequence not shown), the fusion is still insoluble, but it was observed that high concentrations of urea were effective in solubilizing the protein, more so than in the case of Entry

**Table 2. Peptide sequences for tails attached to GFP using the DNA in Table 1.**

| Entry | Sequence | Fusion property |
|---|---|---|
| 1 | WFFHHYYWWWHHHFFWDEDE | insoluble |
| 2 | WFFHEHYYWEWWHHEHFFWE | insoluble |
| 3 | WFFDEHYYDEWWHDEHFFDE | soluble |
| 4 | WWWDEWWWDEWWWDEWWWDE | insoluble |
| 5 | GGGGGGYGPIPFGRGHGHGQ | high IMAC affinity |
| 6 | GGGGGGYGPAPFGRGHGHGQ | high IMAC affinity |
| 7 | GGGGGGGHYSGNPPPGSGWH | medium IMAC affinity |
| 8 | GGGGGGGHYSGSRWH | high IMAC affinity |

4. It was not determined if the "insoluble" GFP fusions were in fact inclusion bodies [13], but the fact that strong fluorescence is observed for all the tails shown in Table 2 suggests that at the least, protein aggregation does not have a strong effect on GFP fluorescence, as has been observed when GFP fusions are found in inclusion bodies [14]. Variations of Entry 3, all with the same placement of DE but with different random aromatic residues, displayed good solubility (five in all, sequences not shown).

Entries 5 and 6 illustrate the ease with which a mutation can be made which involves changing only two of the five segments: the final ATT of 5a is changed to GCG in 6a for the isoleucine to alanine mutation. While segments 5a and "5 tie" are changed, segments 5b, "5a reverse," and "5b reverse" remain unchanged (see Fig 1). Once DNA segments are in hand, and they do not code for a mutation, they can be reused in other experiments, with obvious cost savings. Entries 5 and 6 were an attempt to design a peptide which could interact strongly with DNA, and while this experiment proved unpromising, the HGH sequence near the end of the tail gave the fusion high IMAC affinity (Fig 2), which demonstrates that the short HGH sequence may work as well as the longer His$_6$ sequence. For shorter peptides, there is an advantage to conserving amino acids with the shorter affinity tag, and not "wasting" them with the longer His$_6$. There is also the possibility the sequence HGH itself could be incorporated into a peptide fold, in which case this tripeptide motif would be dual-purpose: one for intramolecular binding and folding, and one for imparting high IMAC affinity to the peptide. Entries 5 and 6 are examples where the chemical properties of the tail all but confirm a successful DNA insertion experiment.

Entry 7 is a design with a series of three proline residues surrounded by various amino acids which showed promise as a folded peptide using the molecular mechanics software Avogadro [15]. Details of these modeling efforts will be the subject of a future publication. Since the IMAC affinity was found to be "medium," meaning the protein eluted as fluorescent fractions before the imidazole wash, but did not elute with the dead volume of the column (Fig 2), this was taken as evidence of the design being a partial success. Notably, the IMAC retention got worse over time as a result of the protein being stored. When a mutant was prepared with an Asn→Asp mutation (sequence not shown), low IMAC retention was observed, suggesting gradual hydrolysis of the amide residue on asparagine [16] was responsible for the deterioration of the IMAC affinity observed for Entry 7. As before, the ease of introducing a mutation makes this test relatively easy and fast. Entry 8 DNA shows another attempted mutation, which was an effort to improve the IMAC affinity of peptide 7 by changing asparagine to serine. However, an insertion error occurred, as described below.

The Entry 8 peptide is relatively short with two histidine residues separated by six amino acids. The fusion has high IMAC affinity (Fig 2), which suggests the intervening amino acids fold to bring the histidine residues into close proximity such that they chelate a metal effectively. The DNA for Entry 8 does not correspond to this peptide. It was found that in this experiment, an insertion error occurred, which was confirmed by sequencing the DNA. The two DNA strands, the designed insert, and the DNA that emerged after the error, can be compared in Table 3. The insertion error resulted in remarkably specific changes to the DNA. In the first place, much of the DNA was unchanged, but a mutation occurred (shown in bold) where a proline codon was changed to an arginine codon. Second, the underlined portion of the insert was deleted. And finally, the remaining DNA was unchanged and remained in frame such that the stop codon was read terminating the peptide. These details are included to describe the error, and are not meant to explain the undoubtedly complex processes that take place *in vivo* as the nicked DNA is replicated. Of the over 200 insertion experiments performed, two insertion errors were observed, the one shown in Table 3, and another whose details are not included and whose DNA design was unrelated to Entries 7 or 8. Thus, for

**Table 3. Comparison of entry 8 to the actual DNA that resulted after the insertion error.**

| DNA | Sequence |
|---|---|
| Entry 8 | CCGAGCTCGGAGGTGGAGGTGGAGGTGGACATTATAGC |
| Actual | GAGCTCGGAGGTGGAGGTGGAGGTGGACATTATAGC |
| Entry 8 | GGCAGC**CCG**CCGCCGGGCAGCGGCTGGCATTAACTCTAGAGG |
| Actual | GGCAGC**CGC** TGGCATTAACTCTAGA |

plasmids sequenced using the colony selection procedure outlined, the five segment insertion technique described here has an error rate estimated to be close to 1%. It should be emphasized, however, the true insertion error rate would be found by sequencing all plasmids from all colonies, and could include a study of which DNA sequences are more prone to errors. Such a study is beyond the scope of this work. The 1% figure used here is meant to suggest a low but still significant error rate, as a caution to researchers contemplating the use of this technique. Additionally, it is unknown how the method will perform with proteins other than GFP, with its convenient green fluorescent phenotype. It is notable and fortunate that the "medium" IMAC affinity was indeed improved in going from Entry 7 to Entry 8, and that the insertion error fortuitously points to a short peptide with good folding characteristics. An insert coding for the Entry 8 peptide (not shown) gives a fusion product with the same high IMAC affinity. The details of this peptide and how it folds are the subject of ongoing research efforts.

Aspects of this methodology that should be highlighted are the 20 nt overlaps (Fig 1) which result in stable double stranded DNA. For DNA 20mers, melting temperatures range from 50–80˚C, depending on the GC content (Invitrogen data, 50 mM $Na^+$). These temperatures are well above the heat shock temperature of 42˚C. The stability of the double stranded DNA could of course be an additional design parameter, where codons with higher GC content could be favored. In this work, generally codons were selected to be most easily expressed in *E. coli*, with no attention paid to GC content. Given the high stability of the five-segment inserts, it seemed possible the idea could be extended to seven segments, where now the number of phosphate-free nicks is five. Unfortunately, no successful transformations with inserts having more than three nicks were observed, suggesting there is a limit to the amount of repair the cells can accomplish. The implication is that overlapping segments with phosphate-free nicks cannot be used to build very large DNA sequences, but longer DNA could be made by lengthening "a" and "b" in Fig 1. An additional potential application of the methodology (pointed out by a reviewer) is the insertion of these cassettes into internal portions of genes coding for proteins, whereby internal protein sequences can be added or mutated.

## Conclusion

This work describes a method for economically constructing DNA cassettes from five oligomers 40 nt or less in length. The low cost of the DNA oligomers, coupled with the restriction digests and molecular biology experiments described, make it possible to investigate multiple mutations rapidly and conveniently in long (up to 20 amino acid) peptides. The 20 nt overlaps allow for the construction of stable double stranded DNA cassettes with three phosphate free nicks. The insertions are generally successful, with an error rate of about 1%. Larger constructs, with five or more nicks, do not lead to successful insertions. A number of peptides, as GFP fusions, are described which elucidate chemical properties such as solubility and IMAC affinity. It is hoped this technique opens up new pathways in peptide research for the discovery of designed functional peptides which can be prepared and characterized quickly and cheaply.

## Author Contributions

**Writing – original draft:** Jonathan Filley.

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
