## [Decision Letter · Decision Letter 0]

31 May 2024

PONE-D-24-14914A convenient five-segment cassette procedure for DNA insertions coding for novel peptidesPLOS ONE

Dear Dr. Filley,

Thank you for submitting your manuscript to PLOS ONE. After careful consideration, we feel that it has merit but does not fully meet PLOS ONE’s publication criteria as it currently stands. Therefore, we invite you to submit a revised version of the manuscript that addresses the points raised during the review process.

We look forward to receiving your revised manuscript.

Kind regards,

Dhana Govind Gorasia

Academic Editor

PLOS ONE

2. We noted in your submission details that a portion of your manuscript may have been presented or published elsewhere. [Figure 1 was presented at an ACS meeting in 2020, but few details of the insertion procedure were given at that time] Please clarify whether this [conference proceeding or publication] was peer-reviewed and formally published. If this work was previously peer-reviewed and published, in the cover letter please provide the reason that this work does not constitute dual publication and should be included in the current manuscript.

Additional Editor Comments:

Reviewer #2 has raised valid concerns which need to be addressed. Also, in the letter to the editor, the authors have mentioned that they are preparing another manuscript with all the results going in that second manuscript. This manuscript can only proceed if the figures requested by reviewer #2 are included in the current manuscript.

Reviewers' comments:

Reviewer's Responses to Questions

**Comments to the Author**

1. Is the manuscript technically sound, and do the data support the conclusions?

Reviewer #1: Yes

Reviewer #2: Partly

2. Has the statistical analysis been performed appropriately and rigorously? 

Reviewer #1: N/A

Reviewer #2: N/A

3. Have the authors made all data underlying the findings in their manuscript fully available?

Reviewer #1: Yes

Reviewer #2: Yes

4. Is the manuscript presented in an intelligible fashion and written in standard English?

Reviewer #1: Yes

Reviewer #2: Yes

5. Review Comments to the Author

Reviewer #1: This study describes a method for the convenient access to peptide mutation libraries with particular consideration to the economic assembly of DNA cassettes. The protocols appear to have been well executed, the advantages of the five segment insertion method demonstrated, and the structure/properties of the resultant peptides considered in the discussion. The authors have also appropriately discussed the scope and limitations of the approached described in the text.

Reviewer #2: This manuscript describes used of short oligonucleotides to make hybrid cassettes for ligation to the 3’ of a GFP-encoding gene in a plasmid. The effect of various fusions on GFP- peptide fusion product fluorescence, solubility and affinity for nickel-chelate resin is described although no experimental data such as gels showing solubility/insolubility of GFP to support the conclusion is presented. The conclusion is that use of short overlapping oligonucleotides can expedite production of recombinant peptides with targeted mutation and be a cost and time-effective alternative to peptide synthesis or use of longer intact DNAs as plasmid inserts.

The paper describes only one protein, GFP as fusion partner with the coded peptides. Thus, it is a study of the effect of changing C-terminal peptide sequences on GFP. The cassette approach is claimed to be cost effective relative to purchase of short peptides, but with a cost of $60 for a peptide (dependent on locale) the cost of buying the oligonucleotides, cloning, harvesting plasmids, sequencing and then examining the protein produced would be similarly expensive and not that much shorter in time frame. The advantage of the method is the fact that the fused protein is large, which would be prohibitive to synthesis. This should be emphasized. Also, the method could viably be used for internal sequence replacement in a gene coding a protein of interest.

Abstract: Describes the work adequately.

Introduction: Sufficient.

Method: I have questions regarding methods. To ensure obtained sequence changes are as expected, do the oligonucleotides used require purification eg by HPLC or just desalting?

The author refers to denaturation of the restriction endonuclease. How is this achieved? If by heating, which is the standard practice, how does this affect the insert DNAs? If by addition of chemicals, how is it ensured that ligation is not inhibited?

If there is a small change to replace for example a single nucleotide for an amino acid substitution, is it necessary to have the reverse complements strand with this change? I would presume this would give a 50:50 rate of wild type vs mutated sequence in retrieved plasmids. Was this tested?

Has the author used degenerate oligonucleotides to produce mutated peptides? Please comment here or in discussion.

Results and discussion: Page 13:

with respect to the conclusion that “close to eight polar residues are required for good solubility when designing 20 amino acid tails.” Since GFP was the only protein tested as a fusion, the statement should be more conservative and state that “close to eight polar residues are required for good solubility when designing 20 amino acid tails appended to GFP”.

Were any analyses performed to determine what proportion of ALL colonies, brightly fluorescent or otherwise, contained plasmids with synthetic DNA inserts? If so, what is the nucleotide sequence error rate for ALL plasmids with inserts?

Page 14:

“Thus, the five segment insertion technique described here has an error rate estimated to be close to 1%.” Because the error rate for ALL plasmids produce was not reported, this should read “Thus, for recombinants with bright GFP fluorescence the five segment insertion technique described here has an error rate estimated to be close to 1%.”

Overall, the results as described (but not presented) show promise for the versatility of the system and proof of principle. However, the versatility for carrier proteins which do not have a clearly discernible phenotype like GFP is unknown. With no indication of the true successful insert retention and so the true percent insert error rate, the usefulness for rapid selection of recombinant fusion genes is also not known. This should be made clear in this short report.

6. PLOS authors have the option to publish the peer review history of their article (what does this mean?). If published, this will include your full peer review and any attached files.

Reviewer #1: No

Reviewer #2: No

---

## [Author Response · Author response to Decision Letter 0]

29 Jun 2024

Reviewer #2 has raised valid concerns which need to be addressed. Also, in the letter to the editor, the authors have mentioned that they are preparing another manuscript with all the results going in that second manuscript. This manuscript can only proceed if the figures requested by reviewer #2 are included in the current manuscript.

Response: The concerns of Reviewer #2 are addressed below. “Another manuscript” mentioned above is not in preparation. The future work referred to in the cover letter is yet to be determined, but will include the protocols used in the current manuscript. Thus, the current manuscript (if accepted) will be cited in the “Experimental” section of the future work. The author was unable to locate in the reviewer’s comments any mention of requested figures by Reviewer #2.

Review Comments to the Author (from Reviewer #2)

a. The effect of various fusions on GFP- peptide fusion product fluorescence, solubility and affinity for nickel-chelate resin is described although no experimental data such as gels showing solubility/insolubility of GFP to support the conclusion is presented.

Response: The criterion for solubility of the fusions is the observation of green fluorescence from either the supernatant (soluble) or the pellet (insoluble). A sentence clarifying this has been added in the revised manuscript. Also, the elution profiles from IMAC runs (Fig2) are included as data supporting the “Fusion property” column in Table 2.

b. The cassette approach is claimed to be cost effective relative to purchase of short peptides, but with a cost of $60 for a peptide (dependent on locale) the cost of buying the oligonucleotides, cloning, harvesting plasmids, sequencing and then examining the protein produced would be similarly expensive and not that much shorter in time frame. The advantage of the method is the fact that the fused protein is large, which would be prohibitive to synthesis. This should be emphasized. Also, the method could viably be used for internal sequence replacement in a gene coding a protein of interest.

Response: The author is indebted to the reviewer for pointing out that longer peptides as well as internal sequence replacements can be made in a cost effective manner. The fact the inserts can accommodate 20 amino acid peptides is now emphasized in the Introduction, as well as in the Conclusion. Additionally, a final sentence is included in the Results and Discussion adding the application of the insertion method to creating internal sequence changes (acknowledging the reviewer).

c. To ensure obtained sequence changes are as expected, do the oligonucleotides used require purification eg by HPLC or just desalting?

Response: Desalt purity oligonucleotides are used as received from Invitrogen. This is now stated explicitly in the Materials and Methods.

d. The author refers to denaturation of the restriction endonuclease. How is this achieved? If by heating, which is the standard practice, how does this affect the insert DNAs? If by addition of chemicals, how is it ensured that ligation is not inhibited?

Response: The restriction enzymes are denatured by heating, which is now stated explicitly in the Materials and Methods. Since the insertions are generally successful, it is assumed that heating the DNA does not irreversibly damage it. No additional chemicals which might interfere with ligation are introduced.

e. If there is a small change to replace for example a single nucleotide for an amino acid substitution, is it necessary to have the reverse complements strand with this change? I would presume this would give a 50:50 rate of wild type vs mutated sequence in retrieved plasmids. Was this tested?

Response: Every mutation introduced in the 5’-to-3’ direction was accompanied by its reverse compliment. This is illustrated for Entries 5 and 6 in Table 1 and in the discussion of these entries. Attempting a mutation by changing only the 5’-to-3’ DNA was never tried. 

f. Has the author used degenerate oligonucleotides to produce mutated peptides? Please comment here or in discussion.

Response: From the manuscript near the end of Results and Discussion: “In this work, generally codons were selected to be most easily expressed in E. coli, with no attention paid to GC content.” This sentence offers the main criterion used for DNA design, namely, that it is most easily expressed in E. coli. Once a DNA sequence was found to work, it was used again. As an example, the DNA coding for the leading poly Gly of Entry 5 is GGA GGT GGA GGT GGA GGT where the alternating codons were picked in order to remove excessive repeats in the DNA (and with no change in Tm). No experiments were done to see if alternate DNA coding for (Gly)6 would perform better, and no mutations were attempted with two sets of degenerate DNA.

g. with respect to the conclusion that “close to eight polar residues are required for good solubility when designing 20 amino acid tails.” Since GFP was the only protein tested as a fusion, the statement should be more conservative and state that “close to eight polar residues are required for good solubility when designing 20 amino acid tails appended to GFP”.

Response: The change has been made, and also includes the more precise “carboxylate” over the less accurate “polar.”

h. Were any analyses performed to determine what proportion of ALL colonies, brightly fluorescent or otherwise, contained plasmids with synthetic DNA inserts? If so, what is the nucleotide sequence error rate for ALL plasmids with inserts?

Response: No.

i. “Thus, the five segment insertion technique described here has an error rate estimated to be close to 1%.” Because the error rate for ALL plasmids produce was not reported, this should read “Thus, for recombinants with bright GFP fluorescence the five segment insertion technique described here has an error rate estimated to be close to 1%.”

Response: The change has been made, but using the following, which is stylistically closer to the manuscript: “Thus, for plasmids sequenced using the colony selection procedure outlined, the five segment insertion technique described here has an error rate estimated to be close to 1%.”

j. Overall, the results as described (but not presented) show promise for the versatility of the system and proof of principle. However, the versatility for carrier proteins which do not have a clearly discernible phenotype like GFP is unknown. With no indication of the true successful insert retention and so the true percent insert error rate, the usefulness for rapid selection of recombinant fusion genes is also not known. This should be made clear in this short report.

Response: The following passage is included after the 1% error rate sentence: “It should be emphasized the true insertion error rate would be found by sequencing all plasmids from all colonies, and could include a study of which DNA sequences are more prone to errors. Such a study is beyond the scope of this work. The 1% figure used here is meant to suggest a low but still significant error rate, as a caution to researchers contemplating the use of this technique. Additionally, it is unknown how the method will perform with proteins other than GFP, with its convenient green fluorescent phenotype.”

---

## [Editor Report · Decision Letter 1]

10 Jul 2024

A convenient five-segment cassette procedure for DNA insertions coding for novel peptides

PONE-D-24-14914R1

Dear Dr. Filley,

We’re pleased to inform you that your manuscript has been judged scientifically suitable for publication and will be formally accepted for publication once it meets all outstanding technical requirements.

Kind regards,

Dhana Govind Gorasia

Academic Editor

PLOS ONE
---

## [Editor Report · Acceptance letter]

17 Jul 2024

PONE-D-24-14914R1 

PLOS ONE

Dear Dr. Filley, 

I'm pleased to inform you that your manuscript has been deemed suitable for publication in PLOS ONE. Congratulations! Your manuscript is now being handed over to our production team.

Kind regards, 

on behalf of

Dr. Dhana Govind Gorasia 

Academic Editor

PLOS ONE